# Neural Brain Fields:
# A NeRF-Inspired Approach for Generating Nonexistent EEG Electrodes

## Abstract

Electroencephalography (EEG) data present unique modeling challenges because recordings vary in length, exhibit very low signal to noise ratios, differ significantly across participants, drift over time within sessions, and are rarely available in large and clean datasets. Consequently, developing deep learning methods that can effectively process EEG signals remains an open and important research problem. To tackle this problem, this work presents a new method inspired by Neural Radiance Fields (NeRF Mildenhall et al. (2021)). In computer vision, NeRF techniques train a neural network to memorize the appearance of a 3D scene and then uses its learned parameters to render and edit the scene from any viewpoint. We draw an analogy between the discrete images captured from different viewpoints used to learn a continuous 3D scene in NeRF, and EEG electrodes positioned at different locations on the scalp, which are used to infer the underlying representation of continuous neural activity. Building on this connection, we show that a neural network can be trained on a single EEG sample in a NeRF style manner to produce a fixed size and informative weight vector that encodes the entire signal. Moreover, via this representation we can render the EEG signal at previously unseen time steps and spatial electrode positions. We demonstrate that this approach enables continuous visualization of brain activity at any desired resolution, including ultra high resolution, and reconstruction of raw EEG signals. Finally, our empirical analysis shows that this method can effectively simulate nonexistent electrodes data in EEG recordings, allowing the reconstructed signal to be fed into standard EEG processing networks to improve performance.

## 1 Introduction

Electroencephalography (EEG) records millisecond scale voltage fluctuations generated by groups of brain cells and remains the most widely used non-invasive technique for tracking real-time brain activity. Its scientific and societal value rests on three pillars: (i) it enables real-time brain technology interaction through brain-computer interfaces and neurofeedback systems (Yun, 2024), (ii) it serves as a clinical gold standard for diagnosing and monitoring neurological conditions such as epilepsy, sleep disorders, and coma, and (iii) it provides a high temporal resolution discovery tool for cognitive and systems neuroscience, illuminating processes such as perception, attention, and memory through event related potentials and oscillatory analyses.

EEG signal processing is inherently difficult due to its low signal to noise ratio, which is susceptible to various artifacts (Niedermeyer & da Silva, 2005), and its limited spatial resolution caused by volume conduction (Sanei & Chambers, 2013). This spatial limitation arises from sparse sensor placement and the smearing of signals across the scalp (Nunez & Srinivasan, 2006), constituting a core constraint of EEG despite its widespread use and excellent temporal resolution. As a result, the signals captured at the scalp are coarse, spatially blurred reflections of the underlying cortical dynamics, with the limited electrode count often insufficient to resolve the true complexity of neural activity patterns.

Early EEG pipelines relied on band-pass filtering, Independent Component Analysis, and hand crafted features such as power spectra or common spatial patterns, which were then fed to shallow classifiers. Over the past decade, deep learning has transformed EEG research. Convolutional

neural networks extract spectrotemporal patterns directly from raw signals (Schirrmeister et al., 2017; Lawhern et al., 2018b; Svantesson et al., 2021a), recurrent models capture temporal dynamics (Ruffini et al., 2016; Roy et al., 2019), graph neural networks leverage the spatial topology of the electrode array (Klepl et al., 2024; Demir et al., 2021), and transformer architectures combine attention mechanisms with self supervised pre-training to improve cross subject generalization (Wang et al., 2024). Together, these approaches now define the SoTA in many EEG tasks such as classification, super resolution, denoising, and others.

Yet, despite steady gains, EEG research has not achieved breakthroughs. Two major bottlenecks continue to hinder comparable advances: First, EEG datasets are limited in both size and spatial coverage, leaving much intracranial brain activity unrecorded. Second, recordings vary across participants, sessions, hardware setups, and environments, and this heterogeneity hinders cross sample and cross dataset generalization. To overcome these bottlenecks, inspired by NeRFMildenhall et al. (2021), we introduce Neural Brain Fields (NBF), a deep learning framework that implicitly learns the underlying dynamics of each recording by mapping a continuous four dimensional space and time field to voltage values. Once trained, the model can estimate voltage at any intracranial location and time point, including regions where no physical electrodes are present, and directly provides tools for reconstruction and super resolution. Our method draws inspiration from techniques in the domain of 3-D scene representation, which use a set of 2-D images to encode the scene into neural weights by learning a continuous function over space and viewing direction, thus enabling scene rendering from arbitrary viewpoints.

The core idea is to overcome the challenges of cross sample generalization caused by non uniform EEG representations by learning the underlying representation of each sample separately, rather than learning shared representations across samples. Moreover, we hypothesize that the remarkable sample efficiency of NeRF based techniques can help mitigate the data constraints commonly found in the EEG domain by producing less noisy signals and supporting high resolution reconstructions, including at locations where no physical electrodes are present and direct access is not possible.

**Our main contributions** encompass the following three aspects: (i) We propose Neural Brain Fields (NBF), a NeRF like approach designed to learn a latent representation of individual EEG recordings, which can be used after training to render voltage signals at arbitrary spatial and temporal locations. (ii) We empirically demonstrate that, with appropriate architectural choices, such as suitable positional encoding and effective normalization, NBF can accurately predict voltages at non existent electrode positions. This capability amplifies the EEG signal and improves the performance of downstream neural networks on tasks such as decoding speech perception from brain activity, emotion classification, and a broad range of other cognitive and affective state decoding tasks. (iii) Our technique is broadly applicable and provides a set of tools for a wide range of EEG processing tasks, including super resolution, signal reconstruction, learning fixed size and continuous representations through model weights, and producing high quality, smooth visualizations.

## 2 BACKGROUND AND RELATED WORKS

**Neural Radiance Fields**  NeRF (Mildenhall et al., 2021)) is a widely used technique in computer vision for learning 3D scene representations, enabling both photorealistic rendering and scene manipulation. Given a set of posed images of a single scene denoted by $s$, NeRF constructs a Ray sample dataset by casting rays through image pixels and sampling 3D points along each ray. This dataset can be formally defined as:

$$\mathcal{D}_s = \left\{ \left( \mathbf{x}_\ell^{(s)}, \theta_s, \phi_s, y_s \right) \,\middle|\, \ell = 1, \ldots, L \right\}, \quad \mathbf{x}_\ell^{(s)} \in \mathbb{R}^3, \theta_s \in [0, 2\pi), \phi_s \in [0, \pi], y_s \in \mathbb{R}^3 \quad (1)$$

where x,y,z are physical 3D coordinates in the visual space and $\Theta$ and $\Phi$ denote the azimuthal and polar angles, respectively. Using this dataset, NeRF learns a continuous volumetric scene representation via a neural function $f_w$, typically implemented as a simple MLP. The function maps each 3D location $\mathbf{x} \in \mathbb{R}^3$ and viewing direction $(\theta, \Phi)$ to a color and volume density:

$$f_w(\mathbf{x}, \mathbf{d}) \mapsto (\mathbf{c}, \sigma), \quad \mathbf{c} \in \mathbb{R}^3, \sigma \in \mathbb{R}_{\geq 0} \quad (2)$$

The predicted outputs are aggregated along rays using differentiable volumetric rendering, resulting in a predicted color for each data point. The rendered color $\hat{\mathbf{C}}(\mathbf{r})$ is then compared to the ground truth color $y_s$, and the network is trained to reconstruct the observed pixel color. Through this process, the network implicitly learns to encode both the geometry and appearance of the scene within its

weights. Once the network is trained, it can render the scene from novel viewpoints by querying the learned function. This rendering process enables photorealistic novel view synthesis and has been applied to a wide range of tasks, including 3D reconstruction, relighting, scene editing (Yuan et al., 2022), virtual reality, and content creation for films and games. Inspired by this line of work, our approach aims to learn the underlying dynamics of individual EEG signals to obtain compact, fixed size neural representations encoded directly in the network's weights. Additionally, we hypothesize that advanced NeRF based rendering techniques such as super resolution, nonexistent electrode synthesis, and EEG signal reconstruction can play a crucial role in amplifying the underlying sources of EEG activity. These techniques may help mitigate data inefficiency and noise, two major challenges in EEG modeling.

**DL-based EEG Processing**  In recent years, many DL methods have been proposed to improve EEG analysis, classification, and generation. This line of work spans a wide spectrum from simple, shallow networks to large scale architectures based on pre-trained Transformers, such as EEGPT (Wang et al., 2024). Notable examples include: (i) CNN-based methods (Schirrmeister et al., 2017; Lawhern et al., 2018b; Svantesson et al., 2021a), (ii) RNN-based methods (Tan et al., 2020), (iii) GNN-based methods (Klepl et al., 2024; Demir et al., 2021), and (iv) Transformer models (Wan et al., 2023; Wang et al., 2024), and (iv) modern architectures such as Mamba-based models (Gui et al., 2024; Yang & Jia, 2024) and diffusion models (Lopez et al., 2025). Moreover, adaptation of DL tools such as design dedicated data augmentation and positional encoding techniques have also been explored for EEG (Rommel et al., 2022; Torma & Szegletes, 2025; Jia et al., 2024). However, rather than investigating which architectures can improve classification accuracy or cross sample generalization, we focus on a less explored research direction: Investigate whether it is possible to learn the underlying source of a single EEG sample, and use this insight to build robust EEG representations that enable signal amplification via rendering techniques.

**Relevant Advances in NeRF and Neural EEG Modeling**  Although our work focuses on adapting NeRF style representations to EEG data, it builds on several recent developments in neural rendering. Mip-NeRF (Barron et al., 2021) and Mip-NeRF 360 (Barron et al., 2022) introduced multiscale sampling and anti-aliasing for unbounded scenes. Instant-NGP (Müller et al., 2022), KiloNeRF (Reiser et al., 2021), FastNeRF (Garbin et al., 2021), and TinyNeRF (Zhao et al., 2023) proposed runtime and architectural optimizations for efficient rendering. Editable NeRF (Yuan et al., 2022) enabled relighting and geometry editing, while Mega-NeRF (Turki et al., 2022) demonstrated large scale scene reconstruction. These advances highlight the scalability and flexibility of neural fields and motivate their adaptation to EEG tasks such as spatial super resolution, signal reconstruction, and virtual electrode generation.

Parallel progress in neural EEG modeling has introduced generative and representation learning methods, including diffusion based approaches have been proposed for EEG denoising (Torma & Szegletes, EasyChair, 2023), and deep networks have been used for EEG to image reconstruction and visual representation learning (Singh et al., 2023; 2024). Several studies have also explored conditioning NeRF style or 3D generative models on EEG signals for object and scene reconstruction (Xiang et al., 2024; Ge et al., 2025). Finally, foundation models such as LUNA leverage NeRF inspired positional encodings for robust cross subject decoding (Döner et al.).

Unlike prior approaches that rely on generative models or representation learning for tasks such as denoising and classification, our method directly models the continuous spatial structure of EEG voltage across the scalp using neural fields.

**EEG Signal Super Resolution**  In the context of EEG, super resolution refers to the task of reconstructing or synthesizing EEG signals at higher spatial resolution than what was originally recorded, typically by estimating signals at unmeasured electrode locations. Several recent studies have explored learnable approaches for enhancing EEG spatial resolution, including the generation of signals at unmeasured or missing electrode positions. Notably, Svantesson et al. (2021b) introduced *Virtual EEG-electrodes: Convolutional neural networks as a method for upsampling or restoring channels*, which proposed CNN based mappings to restore high density EEG from reduced montages (e.g., 4→21, 14→21, or 20→21 electrodes). While their work is an important contribution toward EEG upsampling, it relies on fixed mappings defined for a specific dataset and does not generalize beyond the predefined input output configurations. Similarly, the work *Deep EEG super-resolution: Up sampling EEG spatial resolution with Generative Adversarial Networks* Corley & Huang (2018) applied GANs to EEG super resolution on a limited dataset with constrained input

output mappings. Although promising, the authors note that training GANs for EEG data was computationally intensive and challenging to optimize, reflecting the inherent complexity of learning realistic signal distributions across different subjects and electrode locations.

In contrast, our method is designed for per subject and per recording modeling, allowing the generation of EEG signals at arbitrary spatial locations without being restricted to fixed channel to channel mappings. This flexibility enables the synthesis of electrodes across the entire scalp topology, adapting to the substantial variability in individual head shapes and signal characteristics. By leveraging neural implicit representations, our approach learns a continuous spatial field of EEG activity that generalizes to unseen positions during inference, offering a subject and record specific alternative for EEG signal super resolution.

## 3 METHOD

We begin by defining the learning problem and its motivation in Sec. 3.1. We then present the architecture designed to solve this problem in Sec. 3.2. Finally, we describe how the learned per-sample network can be applied to multiple rendering tasks in Sec. 3.3.

### 3.1 LEARNING PROBLEM

To learn the underlying latent representation of EEG, we introduce a NeRF-inspired DL architecture specifically tailored to model spatiotemporal EEG dynamics. The objective is to approximate a continuous function $f : \mathbb{R}^4 \to \mathbb{R}$, which maps a four dimensional vector comprising three spatial coordinates $(x, y, z)$ and one temporal coordinate $t$ to a scalar voltage value $v$: $(x, y, z, t) \mapsto v$.

Motivated by NeRF and the inherent difficulty of cross-sample generalization in EEG, where signal characteristics vary widely across participants, sessions, and recording conditions, we fit a separate network for each sample. This approach circumvents the variability introduced by such non-uniform representations and instead focuses on faithfully capturing the latent structure of individual recordings, leading to more robust and accurate reconstructions.

**Input Normalization** To ensure numerical stability and consistent scaling across spatial and temporal dimensions, we apply min-max normalization to the coordinate vector $(x, y, z, t)$. The spatial coordinates are normalized jointly using their minimum and maximum values, while the temporal dimension is normalized independently:

$$(x', y', z', t') = \left( 2\, \frac{x - s_{\min}}{s_{\max} - s_{\min}} - 1,\ 2\, \frac{y - s_{\min}}{s_{\max} - s_{\min}} - 1,\ 2\, \frac{z - s_{\min}}{s_{\max} - s_{\min}} - 1,\ \frac{t - t_{\min}}{t_{\max} - t_{\min}} \right), \quad (3)$$

where $s_{\min} = \min\{x, y, z\}$, and $s_{\max} = \max\{x, y, z\}$. The normalized input is denoted as $\mathbf{v}' = (x', y', z', t') \in \mathbb{R}^4$.

**Voltage Normalization and Denormalization** Target voltage values are standardized using z-score normalization inspired by (Apicella et al., 2023):

$$v_{\text{norm}} = \frac{v - \mu}{\sigma}, \quad (4)$$

where $\mu$ and $\sigma$ denote the mean and standard deviation of training voltages. The predicted voltage is denormalized during inference by $\hat{v} = \hat{v}_{\text{norm}} \cdot \sigma + \mu$, ensuring that outputs remain consistent with real world EEG magnitudes.

**Loss Function** To train the model, we employ the Huber loss function Huber (1992), which provides a robust alternative to the Mean Squared Error (MSE). The Huber loss behaves quadratically near the origin and transitions to linear growth for large residuals:

$$\mathcal{L}_\delta(x, y) = \begin{cases} \frac{1}{2}(x - y)^2 & \text{if } |x - y| < \delta \\ \delta \cdot (|x - y| - \frac{1}{2}\delta) & \text{otherwise,} \end{cases} \quad (5)$$

where $\delta$ is a tunable threshold. This formulation is particularly suitable for EEG signals, which may contain occasional transient spikes or artifacts. Unlike the MSE loss, which is highly sensitive to outliers, the Huber loss moderates their influence. This robustness allows the model to focus on learning the underlying neural patterns without being disproportionately affected by noise or corrupted samples that are common in EEG data.

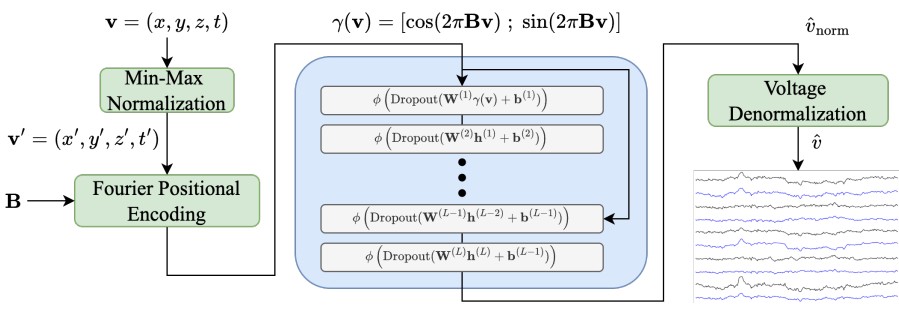

Figure 1: Diagram of the proposed NBF architecture.

## 3.2 ARCHITECTURE

We propose an architecture tailored to capture the continuous spatiotemporal dynamics of individual EEG recordings, with its main components outlined below.

**Fourier Positional Encoding (PE)** To overcome the spectral bias of standard MLPs, which are biased toward learning low frequency functions Rahaman et al. (2019), we employ Fourier-based PE Tancik et al. (2020). This approach maps each normalized input $\mathbf{v} \in \mathbb{R}^4$ to a high-dimensional embedding using sinusoidal functions across multiple frequencies:

$$\gamma(\mathbf{v}) = [\cos(2\pi\mathbf{B}\mathbf{v}) \; ; \; \sin(2\pi\mathbf{B}\mathbf{v})] \in \mathbb{R}^{2m} , \tag{6}$$

where $\mathbf{B} \in \mathbb{R}^{m \times 4}$ is a matrix with entries sampled i.i.d from a Gaussian distribution, $B_{ij} \sim \mathcal{N}(0, \sigma^2)$. The hyperparameter $\sigma$ controls the frequency scale and affects the model's ability to learn high frequency signal components.

**Multi Layer Perceptron (MLP)** The encoded vector $\gamma(\mathbf{v})$ is passed through a fully connected MLP composed of $L$ layers of width $W$, each using ReLU activation, dropout, and periodic skip connections. The network is defined as:

$$\mathbf{h}^{(0)} = \gamma(\mathbf{v}) , \quad \text{for } l = 1, \dots, L-1: \quad \mathbf{h}^{(l)} = \phi\left(\text{Dropout}(\mathbf{W}^{(l)}\mathbf{h}^{(l-1)} + \mathbf{b}^{(l)})\right) , \tag{7}$$

with the output computed as:

$$\hat{v}_{\text{norm}} = \mathbf{W}^{(L)}\mathbf{h}^{(L-1)} + \mathbf{b}^{(L)} . \tag{8}$$

For skip-connected layers, the input $\gamma(\mathbf{v})$ is concatenated with intermediate features:

$$\mathbf{h}^{(l)} = \phi\left(\text{Dropout}(\mathbf{W}^{(l)}[\mathbf{h}^{(l-1)}; \gamma(\mathbf{v})] + \mathbf{b}^{(l)})\right) . \tag{9}$$

**Model Initialization** To accelerate the overall training process, we utilize a progressive fine tuning approach. The model is first trained from scratch on the initial window for a full number of epochs and then saved. For each subsequent window, we initialize training from the previously saved model and fine tune it for a reduced number of epochs, subsequently overwriting the saved checkpoint. This iterative strategy reduces the total training time by approximately $30\%$, while simultaneously improving reconstruction accuracy. The improvement is attributed to the temporal coherence that exists across neighboring windows, which the model exploits when initialized with weights from a temporally adjacent segment.

## 3.3 RENDERING APPLICATIONS

**Inference** Once trained, the model is capable of predicting voltage values at arbitrary spatiotemporal coordinates $(x, y, z, t)$, including positions that were not part of the original training dataset. These target locations can be selected based on task specific requirements. For instance, in speech decoding tasks, one may focus on regions proximal to the auditory cortex, such as those corresponding to electrodes T7 and T8. In cognitive neuroscience experiments that involve executive function or attention, predictions may be concentrated in frontal regions represented by electrodes such as Fz. The model's continuous formulation allows for smooth spatial and temporal interpolation, enabling flexible voltage estimation across a range of configurations.

**Data Driven Electrode Augmentation Strategy** To enhance EEG spatial resolution beyond electrode placement limits, we use a data driven augmentation strategy that synthesizes virtual signals at additional scalp sites. This augments the sensor grid with plausible voltages, improving spatial coverage and enabling comprehensive downstream analyses. The model is trained on temporal windows of length $T$, where the number of windows equals the recording duration divided by $T$. In each window, training uses all available electrodes, after which the model synthesizes voltages at new scalp locations not coinciding with the originals. Repeating this across windows produces high resolution synthetic data that captures both spatial proximity and localized temporal dynamics.

## 4 EXPERIMENTS

**Decoding Speech from EEG** Défossez et al. (2023) developed a model that decodes speech perception from noninvasive brain recordings (MEG and EEG) using contrastive learning, employing a dedicated brain module and a pretrained speech module based on wav2vec 2.0. Top 1 accuracy quantifies the proportion of instances in which the model's highest ranked prediction matches the actual speech segment, while top 10 accuracy reflects the frequency with which the correct segment appears among the ten most probable predictions.

**Experimental Protocol and Training Strategy** To capture subject specific neural dynamics efficiently, we employed progressive fine tuning of one Neural Brain Field (NBF) model per subject using temporally segmented EEG/MEG data. Recordings were split into non overlapping 3s windows. For each subject, training began from scratch on the first window with all electrodes. After training on the first window, the model then predicted voltages at novel auditory cortex electrodes for that window, generating synthetic 3s signals integrated into the data stream. Model parameters were check pointed, reloaded for each subsequent window, and fine tuned for fewer epochs. After each step, predictions at the same novel electrodes sites were again generated and inserted per time window. This iterative scheme progressively adapted weights while preserving subject specific features, capturing localized spatiotemporal dynamics and exploiting temporal coherence. Avoiding reinitialization cut training time by $\sim 30\%$, while per subject training maintained physiological fidelity, yielding realistic and stable scalp wide electrode synthesis.

**Evaluating Functional Benefits via Emotion Recognition** To further assess the practical utility of our NBF model in real world EEG applications, we conducted an experiment on emotion recognition. The goal was to determine whether augmenting the spatial coverage of EEG recordings with virtual electrodes synthesized by our model could improve performance in a representative downstream task. We selected the DREAMER dataset (Katsigiannis & Ramzan, 2018), a widely used benchmark for emotion recognition based on EEG, and adopted a standard classification framework using the EEGNet architecture (Lawhern et al., 2018a). Virtual electrodes were synthesized at central scalp locations commonly associated with affective processing, and the augmented data were used to train the classifier following established protocols (Kukhilava et al., 2025). This experiment allows us to evaluate whether the spatially enriched representations generated by our model offer measurable benefits in functional decoding tasks beyond interpolation accuracy alone.

**Hyperparameters** We optimized the NBF model via an extensive grid search over dropout rate, batch size, depth, width, PE levels, and learning rate, using a 90/10 train–validation split. Validation sets were selected for maximal cortical coverage to ensure robust generalization. The search was conducted independently on all three datasets using subsets of participants, and the best configuration was retained. Final settings for most experiments were dropout 0.1, width 1450, depth 8, and batch size 32, trained with Adam and gradient clipping for stability. For the Gwilliams dataset, a larger batch size of 250 was adopted to match dataset scale; this choice was validated on larger participant subsets, confirming robustness across datasets and configurations.

## 5 RESULTS

### 5.1 ENHANCED DECODING PERFORMANCE VIA ELECTRODE AUGMENTATION STRATEGY

As shown in Table 1, the additional electrode data generated by our NBF model significantly enhanced the performance of the Brainmagick (Défossez et al., 2023) and outperformed Transformer-VAE (Chen et al., 2025) across all datasets and metrics in the task of decoding speech perception from noninvasive brain recordings. On the Brennan dataset (Brennan & Hale, 2019), our model achieves a Top-1 accuracy of 9.64 and a Top-10 accuracy of 36.94, compared to 5.2 and 25.7 for

Table 1: Top-1 and Top-10 accuracy comparison of three methods (Original, VAE, and Ours) across three evaluation benchmarks: Brennan, Gwilliams, and Borderick. Results show that our method consistently improves performance across multiple datasets and evaluation metrics.

| Method / Dataset | Brennan | | Gwilliams | | Borderick | |
|---|---|---|---|---|---|---|
| | Top-1 (↑) | Top-10 (↑) | Top-1 (↑) | Top-10 (↑) | Top-1 (↑) | Top-10 (↑) |
| **Brainmagick (Défossez et al., 2023)** | $5.2 \pm 0.8$ | $25.7 \pm 2.9$ | $41.3 \pm 0.1$ | $70.7 \pm 0.1$ | $5.0 \pm 0.4$ | $17.0 \pm 0.6$ |
| **Transformer-VAE (Chen et al., 2025)** | 4.1 | 26.82 | – | – | – | – |
| **NBF (Our)** | $\mathbf{9.64 \pm 0.33}$ | $\mathbf{36.94 \pm 0.1}$ | $\mathbf{42.1 \pm 0.1}$ | $\mathbf{71.33 \pm 0.1}$ | $\mathbf{7.44 \pm 0.13}$ | $\mathbf{22.68 \pm 0.08}$ |

Brainmagick. On the Gwilliams dataset (Gwilliams et al., 2022), our model reaches 42.1 and 71.33, slightly surpassing Brainmagick's 41.3 and 70.7. On the Broderick dataset, we observe similar improvements. These gains are attributed to our method of augmenting the input space with synthetic electrodes near auditory regions. By enriching the spatial representation, the model gains better access to neural dynamics relevant for speech perception, enabling more accurate decoding.

## 5.2 Impact of Electrode Augmentation on Emotion Recognition Performance

To assess the utility of our NBF model in augmenting spatial EEG information for emotion recognition, we conducted experiments on the DREAMER dataset (Katsigiannis & Ramzan, 2018). We employed our model to synthesize EEG signals at five central electrode sites: Cz, Fz, Pz, C3, and C4. These locations were chosen based on prior studies that emphasize their importance in affective processing and emotion classification (Singh & Singh, 2017; Koelstra et al., 2012). By introducing virtual signals at these strategically selected sites, our model effectively extends spatial coverage in brain regions known to carry emotion discriminative features.

To quantify the contribution of the augmented signals to downstream tasks, we trained the EEGNet architecture (Lawhern et al., 2018a) on the DREAMER dataset (Katsigiannis & Ramzan, 2018), supplemented with our synthesized signals from five additional central electrodes (Cz, Fz, Pz, C3, and C4). The training followed the protocol outlined by Kukhilava et al. (Kukhilava et al., 2025), which aligns with contemporary benchmarking standards in EEG based emotion recognition. As shown in Table 2, the inclusion of virtual electrode signals led to consistent improvements in classification accuracy for both arousal and valence dimensions. These findings highlight the effectiveness of the NBF model in enhancing EEG spatial resolution and suggest its potential to improve performance across diverse downstream EEG decoding tasks, including but not limited to affective computing.

## 5.3 Evaluation of EEG Signal Prediction at Unseen Electrode Positions

To assess the model's ability to predict EEG at unseen electrodes, it was trained on 55 of 60 electrodes and tested on the 5 held out positions, repeating three times with different disjoint validation sets across all Brennan dataset subjects. Evaluation used one third of each subject's recordings to reduce computational cost. Performance was compared

Figure 2: Performance comparison of EEGNet on the DREAMER dataset with and without the addition of virtual electrodes generated by our NBF model. Values are reported as mean ± standard deviation across subjects.

| Model | Valence Accuracy | Arousal Accuracy |
|---|---|---|
| EEGNet | $0.60 \pm 0.08$ | $0.46 \pm 0.09$ |
| EEGNet W. Ours | $\mathbf{0.63 \pm 0.03}$ | $\mathbf{0.49 \pm 0.01}$ |

with Spline Surface Interpolation (SSI) Perrin et al. (1989) and Radial Basis Function (RBF) interpolation (Jäger et al., 2016), using identical train test splits and preprocessing. Subjects with extreme baseline errors were excluded; the remaining 30 satisfied SSI's spherical assumptions via MNE-Python's _check_pos_sphere (Gramfort et al., 2013). Negative $R^2$ values were clamped to zero, and extreme SNR outliers discarded. Evaluation was conducted over 80 non overlapping 3 s windows per subject, with averaged results. Table 2 shows the NBF model consistently outperformed baselines, demonstrating robust reconstruction and interpolation at spatially unseen electrodes.

## 5.4 Model Evaluation on Auditory-Centric Electrode Augmentation

To evaluate the effectiveness of our model in generating meaningful synthetic EEG electrode data, we conducted a comparative analysis using the Brainmagick model on the Brennan dataset. We examined the impact of augmenting the original electrode configuration with either three or five additional electrodes. These electrodes were either selected at random or strategically placed near the auditory cortex. As illustrated in the graphs below (Figures 3a and 3b), the inclusion of audi-

Table 2: Performance comparison between our model and baselines across various metrics calculated using the Scikit-learn library (Pedregosa et al., 2011).

|  | MSE (↓) | MAE (↓) | $R^2$ (↑) | PCC (↑) | SNR (↑) | NMSE (↓) |
|---|---|---|---|---|---|---|
| **RBF** (Jäger et al., 2016) | 2.80e−8 | 1.78e−5 | 0.456 | 0.746 | 3.93 | 0.544 |
| **SSI** (Perrin et al., 1989) | 4.19e−8 | 2.06e−5 | 0.399 | 0.720 | 3.33 | 0.601 |
| **NBF (Ours)** | **1.32e−8** | **1.07e−5** | **0.483** | **0.755** | **4.29** | **0.517** |

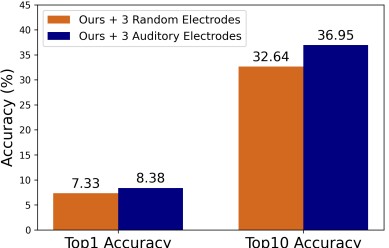

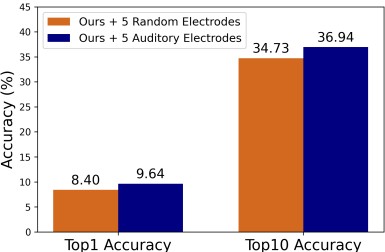

(a) Accuracy with 3 additional electrodes

(b) Accuracy with 5 additional electrodes

Figure 3: Reconstruction accuracy improvement with synthetic auditory electrodes.

tory region electrodes consistently outperformed the addition of randomly placed electrodes in both augmentation settings (three or five electrodes). These findings are consistent with the nature of the task, decoding auditory stimuli, as the auditory cortex is directly involved in speech perception. This suggests that our model has the potential to support task specific electrode augmentation.

## 6 MODEL ANALYSIS

**Reconstructed EEG Signals** To illustrate the model's ability to reconstruct neural activity at spatially unseen sites, we trained it from scratch on a single 3 s window from one subject, using a 90%/10% spatial split of electrodes (training/validation). Training and prediction were restricted to this window and subject, with no leakage from held out electrodes. Figure 4 show representative results: Signal 1 is an MEG channel from the Gwilliams dataset (Gwilliams et al., 2022), and Electrode 2 an EEG channel from the Broderick dataset (Broderick et al., 2018). Time domain plots (4a, 4c) reveal strong alignment between predictions and ground truth, capturing waveform morphology and neural dynamics, while frequency domain plots (4b, 4d) demonstrate spectral coherence across low and high frequencies. These results highlight the model's capacity to synthesize meaningful neural signals at previously unseen spatial locations from limited temporal and spatial data.

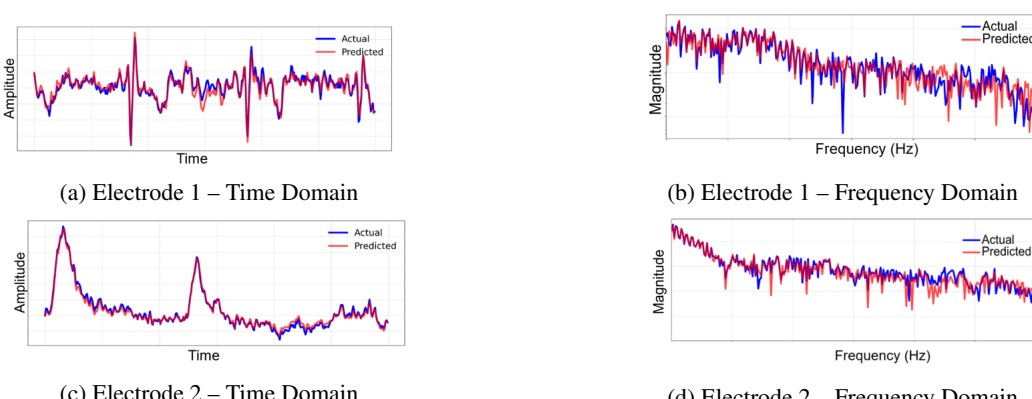

(a) Electrode 1 – Time Domain

(b) Electrode 1 – Frequency Domain

(c) Electrode 2 – Time Domain

(d) Electrode 2 – Frequency Domain

Figure 4: Electrode signals are presented in each row: in (a,c), the time domain; and in (b,d), the frequency domain.

**Qualitative Assessment of Synthesized Electrode Signals** Figure 5 shows synthesized electrode signals for a representative Brennan subject. Each original electrode (black) is paired with its nearest synthesized auditory-related electrode (blue) based on Euclidean distance in the standard montage:

Table 3: Ablation study results for different normalization, encoding, and architectural configurations. Metrics were calculated using Pedregosa et al. (2011).

| Ablation | MSE (↓) | MAE (↓) | R2 (↑) | PCC (↑) | SNR (↑) | NMSE (↓) |
|---|---|---|---|---|---|---|
| **Ours** | **8.18e-11** | **5.20e-06** | **0.680** | **0.8330** | **5.40** | **0.32** |
| Ours – PE (16 levels) | 8.26e-11 | 5.39e-06 | 0.670 | 0.8311 | 5.31 | 0.33 |
| Ours – PE (8 levels) | 8.45e-11 | 5.50e-06 | 0.661 | 0.8300 | 5.32 | 0.34 |
| Ours – w/o Positional Encoding | 1.41e-10 | 7.88e-06 | 0.380 | 0.6360 | 2.64 | 0.62 |
| Ours – MSE loss | 8.32e-11 | 5.59e-06 | 0.657 | 0.8290 | 5.35 | 0.34 |
| Ours – w/o Dropout | 8.38e-11 | 5.54e-06 | 0.657 | 0.8290 | 5.30 | 0.34 |
| Ours – w/o Model Initialization | 8.31e-11 | 5.64e-06 | 0.653 | 0.6526 | 5.20 | 0.35 |
| Ours – w/o Coord. Normalization | 9.01e-11 | 5.65e-06 | 0.630 | 0.8140 | 5.03 | 0.37 |
| Ours – w/o Z-Score Normalization | 2.62e-10 | 1.09e-05 | –0.049 | 0.0300 | 0.04 | 1.05 |
| Ours – w/o Skip Connection | 8.32e-11 | 5.54e-06 | 0.658 | 0.8290 | 5.32 | 0.34 |

59–62, 53–63, 58–64, 54–65, and 60–66. This side by side comparison enables evaluation of temporal structure and morphology. The synthetic signals preserve key waveform patterns and dynamics of the originals, demonstrating physiologically plausible data at novel sites and supporting the model's spatial interpolation validity.

**Ablation Study** As shown in Table 3, we conducted an ablation study on a high quality subset of subjects from the Brennan dataset, using 55 electrodes for training and 5 unseen electrodes for validation. Replacing the Huber loss with MSE slightly reduced performance, confirming Huber's robustness to transient artifacts. Disabling progressive initialization degraded accuracy and increased training time, whereas initializing from adjacent windows reduced training time by ∼30% by leveraging temporal continuity. Removing $Z$ score or coordinate normalization impaired performance, with $Z$ score removal causing catastrophic failure. Similarly, reducing PE depth (e.g., 8 or 16 levels) impaired generalization, while omitting PEs or skip connections

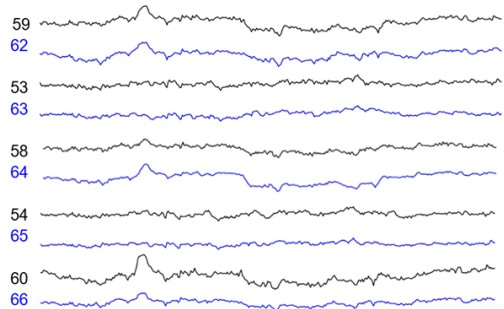

Figure 5: Qualitative comparison of original (black) and synthesized (blue) EEG signals for spatially nearest electrode pairs (59–62, 53–63, 58–64, 54–65, and 60–66) in the Brennan dataset, illustrating preserved waveform morphology and temporal dynamics at novel electrode locations.

substantially reduced accuracy. Overall, each architectural and normalization component proved critical for efficient and accurate EEG reconstruction.

## 7 CONCLUSION

We introduce Neural Brain Fields (NBF), a NeRF-inspired framework for continuous spatiotemporal EEG representations. Trained per subject and recording, NBF generates signals at arbitrary scalp locations, enabling spatial interpolation, synthetic electrode placement, and adaptation to individual head variability. NBF outperforms SSI and RBF interpolation, improves speech decoding in Brainmagick (Défossez et al., 2023) with auditory cortex electrodes, and enhances EEGNet (Lawhern et al., 2018a) emotion classification on DREAMER (Katsigiannis & Ramzan, 2018) via central virtual electrodes, demonstrating generalizable spatial resolution gains. Formally, NBF predicts voltages $v$ from spatiotemporal inputs $(x, y, z, t)$, establishing a flexible framework for other voltage-mapped domains (e.g., geophysics, environmental monitoring) requiring dense spatial interpolation. Current computational cost limits real-time use; however, runtime acceleration is feasible via NeRF advances: multiresolution hash encoding and CUDA optimization (Instant-NGP (Müller et al., 2022)), architectural simplification (tinyNeRF (Zhao et al., 2023)), parallel MLP inference (KiloNeRF (Reiser et al., 2021)), modular computation separation (FastNeRF (Garbin et al., 2021)), and multiscale sampling (Mip-NeRF (Barron et al., 2021)). Adapting these may enable both offline and low latency deployment. Future work will pursue such optimizations and extend NBF to other neuroimaging modalities and downstream applications, translating continuous neural fields into practical neuroscience tools.

## 8 ETHICS STATEMENT

This research utilizing publicly available and requested access EEG datasets commits to maintaining scientific integrity by clearly distinguishing between original and synthetically generated electrode data in all publications, ensuring proper attribution and data licensing compliance, validating synthetic signal quality through rigorous metrics, and acknowledging limitations of the NBF approach. We recognize that synthetic neural signals, despite being derived from legitimate datasets, require transparent reporting to prevent misrepresentation in downstream applications, particularly in clinical or brain computer interface contexts where safety and efficacy depend on authentic neural recordings. The research prioritizes responsible sharing of findings that prevent misuse of synthetic EEG generation technology and adherence to existing institutional ethics frameworks while establishing appropriate boundaries between research applications and potential clinical deployment.

## 9 REPRODUCIBILITY STATEMENT

The reproducibility package provides all necessary instructions and scripts to replicate the experiments reported in our paper. It includes environment setup files, preprocessing pipelines for the Brennan2019 dataset, and end to end scripts for training and evaluating the proposed NBF model. Step by step usage is documented, covering data preparation, model execution, electrode augmentation, and integration with downstream baselines such as Brainmagick. By following the provided guidelines, researchers can reproduce the reported results and further explore the effectiveness of the proposed approach.

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

# A    APPENDIX

You may include other additional sections here.

**Datasets & Preprocessing**    We utilized three publicly available neural datasets. Broderick (Broderick et al., 2018) includes EEG recordings from 19 subjects using 128 sensors (19.2 hours total), collected during continuous narrative speech listening. Brennan (Brennan & Hale, 2019) consists of EEG data from 33 participants (60 sensors, 6.7 hours), recorded during passive listening to a 12.4 minute audiobook, aimed at analyzing hierarchical linguistic prediction. Gwilliams (Gwilliams et al., 2022) provides MEG recordings from 27 subjects (208 sensors, 56.2 hours), focused on phoneme sequence encoding during natural speech perception. All datasets were downsampled to 120 Hz and preprocessed with a 0.5 to 59.5 Hz bandpass filter.

The DREAMER dataset was preprocessed with a bandpass filter in the range 0.3–45 Hz to retain relevant neural frequencies, and a 50 Hz notch filter was applied to remove power line interference.

## A.1    COMPLEXITY ANALYSIS

To assess the computational demands of our method, we measured the total processing time for each dataset. The pipeline involved training on sequential non overlapping 3 second windows, generating synthetic electrode data per window, and iterating through the dataset. Processing the Brennan dataset required approximately 62 hours, Broderick around 172 hours, and Gwilliams the largest and most complex approximately 525 hours. These durations reflect the cumulative runtime across all 3 second windows. All experiments were performed on a parallelized setup with 8 NVIDIA RTX 6000 GPUs, allowing the effective per GPU runtime to be estimated by dividing the total time by 8. This parallelization substantially reduced wall clock time and demonstrates the scalability of our approach on multi GPU systems.

**Biophysical Background**    The feasibility of super resolving EEG signals is grounded in well established biophysical properties of volume conduction, which induce spatial smoothness and strong correlations across neighboring electrode sites (Nunez & Srinivasan, 2006; Michel et al., 2004). This inductive prior is precisely what enables super resolution. Theoretical work has long established that scalp EEG is dominated by smooth, spatially diffuse fields resulting from the physics of current propagation through the brain and skull (Nunez & Srinivasan, 2006). Our results empirically validate that these correlations are present and can be effectively leveraged using a spatially continuous representation learned per subject and per recording. Furthermore, with appropriate fine tuning, our model can adapt to new montages and patient specific geometries, offering a flexible and extensible framework for enhancing spatial resolution beyond traditional interpolation methods or fixed supervised learning paradigms.

**Brennan and Gwilliams Datasets (Brennan & Hale, 2019; Gwilliams et al., 2022)**    Figures 6 and 7 show Top-1 and Top-10 accuracy as functions of added electrodes. On the Brennan dataset, accuracy rises with more electrodes, peaking at 5 before declining, indicating that NBF effectively exploits limited spatial information and generalizes well by interpolating neural activity, validating its utility under sensor constraints. On the Gwilliams dataset, gains are smaller but consistent up to 11 electrodes; despite high baseline performance, NBF improves Top-1 by ∼0.8% and Top-10

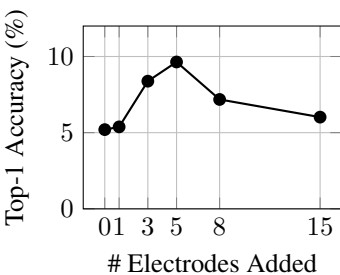 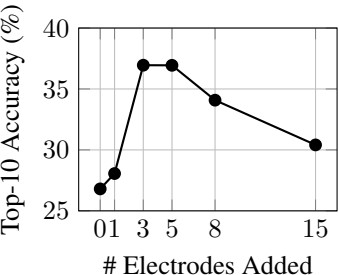

Figure 6: Top-1 and Top-10 accuracy on Brennan dataset as a function of added electrodes.

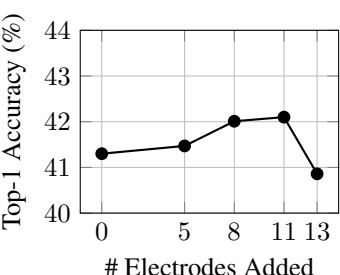 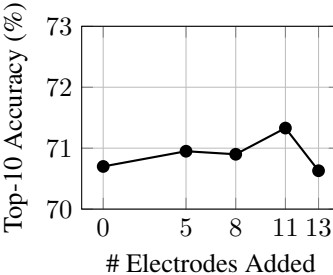

Figure 7: Top-1 and Top-10 accuracy on the Gwilliams dataset as a function of added electrodes.

by ~0.6%, i.e., $8\times$ and $6\times$ the reported 0.1% STD (Gwilliams et al., 2022), strongly surpassing baseline variability and confirming substantial improvement over prior methods.

