# OpenReview forum: "Neural Brain Fields: A NeRF-Inspired Approach for Generating Nonexistent EEG Electrodes"
_ICLR.cc/2026/Conference — Submitted to ICLR 2026_

### Official Review · Reviewer_77z2 · 2025-10-30

**Soundness:** 3
**Presentation:** 2
**Contribution:** 3
**Rating:** 2
**Confidence:** 4

**Summary:**

The paper introduces a novel approach for modeling EEG data by drawing inspiration from Neural Radiance Fields (NeRF). Recognizing the challenges of EEG, such as low signal-to-noise ratio, temporal drift, inter-subject variability, and limited dataset size, the authors propose treating EEG electrodes analogously to camera viewpoints in NeRF, enabling the learning of a continuous representation of neural activity. The proposed network is trained in a NeRF-style manner on a single EEG sample to produce a compact weight vector encoding the entire signal, which can then be used to reconstruct EEG signals at unseen spatial or temporal points. This framework allows for continuous, high-resolution reconstruction of raw EEG signals and accurate reconstruction of missing electrode data, enhancing the performance of downstream EEG task performance.

**Strengths:**

1. The proposed use of implicit neural representations for EEG signal generation and missing-channel reconstruction is both novel and conceptually inspiring.
2. The reconstruction result visualizations are impressive, and the proposed model achieving superior performance compared to the baseline methods.

**Weaknesses:**

Major concerns:
1. The proposed approach is more akin to an implicit neural representation (INR) than to a true Neural Radiance Field (NeRF), as it lacks core NeRF components such as volumetric rendering and per-ray sampling strategies that enable view-dependent reconstruction.

2. Insufficient experimental validation. The experimental evaluation is limited to electrode-missing scenarios and does not explore other important cases, such as missing time points, combined spatiotemporal missing patterns, or varying sparsity levels. As a result, the generality and robustness of the proposed approach remain unverified. Though the idea of this paper is very interesting, a comprehensive evaluation would further largely strenghten the contributions and support the authors’ claims more convincingly.

3. The framework is limited in scalability and practicality. Each EEG sample requires a dedicated per-recording network trained on short (3s) segments with progressive fine-tuning. This setup limits scalability to large cohort of EEG datasets. Probably providing the time for fully finetune for of the model would be helpful.


Minor concerns:
1. The paper should discuss potential limitations when reconstructing signals at boundary electrodes. Similar to NeRF, the proposed model may struggle to extrapolate accurately in regions with no nearby training electrodes, leading to unreliable reconstruction near the scalp edges.

2. The evaluation is conducted on relatively small datasets, which limits the assessment of the model’s scalability and generalization. Including experiments on larger or more heterogeneous EEG datasets would further strengthen the paper’s claims.

3. Some important experimental details are missing or not presented clearly in the main text. For instance, information such as the number of subjects, channel configurations, or which electrodes were masked during evaluation is either hard to find (some of them are in the result section) or only appears in the appendix. These details are essential for understanding and reproducing the experiments and should be summarized clearly in the experiment section.

**Questions:**

1. Although not strictly necessary and somewhat beyond the current scope, it would be interesting to include comparisons with more recent generative models as additional baselines to further strengthen the contribution of this work.
2. How does the model handle reconstruction at boundary electrodes, where spatial extrapolation beyond observed positions is required? Are there any differences in performance between internal vs. boundary electrodes when reconstructing missing channels?

---

> ### Author Response · Authors · 2025-11-24
> **Answers to Questions**
>
> Addressing Questions:
> 1.
> We initially attempted to compare our approach with the recently proposed Spatio Temporal Adaptive Diffusion (STAD) model, which pioneered the application of diffusion models for EEG super resolution on the Localize MI dataset. This dataset comprises high density EEG recordings from seven drug resistant epilepsy patients, acquired using a 256 channel EGI system at 8000Hz. STAD evaluated their method against several deep learning baselines: a convolutional model (DeepCNN), two autoencoder variants and DAE, and a GAN based super resolution method (EEGSR-GAN). The table below presents performance metrics compared to what they reported. While our method demonstrates superior performance across all metrics compared to STAD and the other baseline methods, we lacked access to their specific train/test split and preprocessing details. We contacted the STAD authors to request clarification on their experimental setup, including the exact data segments and train/validation/test partitions, but did not receive a response. Consequently, this comparison was not included in our final evaluation to maintain experimental rigor and reproducibility.
>
>
> **Table 1.** Comparison to prior deep learning based super resolution methods.
>
>
> | Method   | PCC (higher is better) | MAE (lower is better) | NMSE (lower is better) | SNR (higher is better) |
> |---------|------------------------|------------------------|------------------------|------------------------|
> | DeepCNN | $0.20 \pm 0.06$        | $0.36 \pm 0.03$        | $0.45 \pm 0.03$        | $3.00 \pm 0.58$        |
> | WGLAE   | $0.25 \pm 0.06$        | $0.38 \pm 0.03$        | $0.45 \pm 0.03$        | $4.50 \pm 0.29$        |
> | DAE     | $0.225 \pm 0.06$       | $0.375 \pm 0.03$       | $0.45 \pm 0.03$        | $3.75 \pm 0.29$        |
> | EEGSR GAN | $0.245 \pm 0.07$     | $0.35 \pm 0.06$        | $0.425 \pm 0.06$       | $4.00 \pm 0.29$        |
> | STAD    | $0.50 \pm 0.12$        | $0.15 \pm 0.03$        | $0.25 \pm 0.03$        | $8.00 \pm 0.28$        |
> | Ours    | $0.75 \pm 0.06$        | $0.135 \pm 0.06$       | $0.10 \pm 0.06$        | $10.25 \pm 0.43$       |
>
> 2. Our model represents the scalp voltage as a continuous neural field defined over 3D electrode coordinates. Because the representation is continuous, reconstruction does not require explicit extrapolation beyond the observed domain. Instead, boundary electrodes are treated like any other query point, and the positional encoding provides a smooth inductive bias that stabilizes the field near the edges.

---

### Official Review · Reviewer_Drmd · 2025-10-30

**Soundness:** 2
**Presentation:** 3
**Contribution:** 3
**Rating:** 4
**Confidence:** 4

**Summary:**

Inspired by NeRF, this paper models EEG signals as sparse samples from a continuous neural field. Leveraging a MLP architecture, the model reconstructs EEG signals at arbitrary spatial and temporal locations, and serves as an effective data augmentation strategy to enhance downstream task performance.

**Strengths:**

- The transfer of NeRF’s implicit field modeling paradigm to neuroscience is novel.
- Through comprehensive experiments and visualization results, this paper convincingly shows the effectiveness of its modeling approach, which can generate virtual electrode data and improve accuracy on three downstream speech decoding datasets.

**Weaknesses:**

- Current models require training a separate model for each EEG sample, making them impractical for real-world use. More efficient and generalizable solutions are needed.
- The analogy between NeRF and EEG data is conceptually appealing but physically questionable. NeRF samples exhibit spatial continuity and illumination consistency, with multi-view observations providing strong constraints. In contrast, EEG does not represent different views of an implicit field but rather a complex superposition of signals from multiple brain regions via volume conduction. Moreover, EEG/MEG sampling is highly sparse, making it uncertain whether the source current distribution can be effectively learned. The method proposed in the paper appears to be a simple regression model that merely completes voltage values in space.

**Questions:**

- The authors state in both the Abstract and Introduction that the model can predict EEG signals at arbitrary time points(e.g., L028, L081). However, this capability is not demonstrated or evaluated in the experiments. Could the authors clarify or provide supporting evidence?
- In Section 5.1, please specify which electrodes are used as “additional electrodes” in the experiments. Providing this information would improve reproducibility and interpretability.

---

> ### Author Response · Authors · 2025-11-24
> **Answers to Questions**
>
> Addressing Questions:
>
> 1. Our model is defined as a continuous function f(x,y,z,t) so in principle it can be queried at any temporal coordinate inside the 3 second window. This follows directly from the neural field formulation, where time is treated just like the spatial coordinates and is passed through positional encoding and the MLP. Thus, the ability to predict at arbitrary time points is inherent to the representation.
> However, we did not evaluate temporal super resolution in this submission. Our experiments focused exclusively on spatial super resolution, because this is the central practical problem in EEG, the spatial sampling is sparse.
> To avoid confusion, we will revise the manuscript to clarify that the capability of querying arbitrary time points refers to the form of the continuous representation, not to an empirically evaluated temporal super resolution experiment.
>
> 2. In the electrode augmentation experiments of Section 5.1, we generated synthetic signals at a small, fixed set of five electrodes that lie near auditory and speech related cortical areas. For the Brennan dataset, for example, we augmented the input with synthetic activity at five electrodes located near auditory and speech related regions T7, T8, TP7, TP8, and FC5, which cover mid temporal and temporal parietal sites associated with auditory and linguistic processing. These positions were chosen because they typically carry strong task relevant information yet are often sparsely represented in standard low density montages. We will revise the manuscript to explicitly list these electrodes and clarify that the same fixed set was used across all subjects and datasets.

---

### Official Review · Reviewer_hoBu · 2025-10-30

**Soundness:** 3
**Presentation:** 3
**Contribution:** 3
**Rating:** 8
**Confidence:** 4

**Summary:**

This work introduces a novel technique that leverages Neural Radiance Fields (NERF) to enhance the recording of electroencephalogram (EEG) data. By applying this approach, the authors aim to augment the spatial resolution of EEG recordings by increasing the number of electrode positions, thereby capturing a more comprehensive representation of brain activity. Additionally, the technique seeks to improve temporal sensing, allowing for more accurate monitoring of rapid neural dynamics.

**Strengths:**

1. By simultaneously enhancing both spatial and temporal data dimensions, the proposed method exhibits significant potential for improving the accuracy and applicability of EEG studies. This advancement is particularly relevant to fields such as clinical diagnostics, where precise interpretations can lead to better patient outcomes, and neuroscience research, which relies on detailed brain activity monitoring to uncover fundamental neural mechanisms.

2. The study demonstrates notable merit, introducing NeRF as an innovative approach to enhance EEG data quality. This concept stands out not only for its novelty but also for its potential to transform existing analytical frameworks in EEG research and applications.

3. The presentation of the results is clear and well-structured, emphasizing the advantages of the NeRF-based approach. The findings demonstrate that this method outperforms a recently developed alternative utilizing Variational Autoencoders (VAE), highlighting its superiority in processing and interpreting complex EEG data. This comparison adds significant weight to the argument for adopting NeRF techniques in future EEG analysis.

**Weaknesses:**

There is no major weakness evident of this work.

**Questions:**

Nil

**Details Of Ethics Concerns:**

Nil

---

### Official Review · Reviewer_UoCi · 2025-11-02

**Soundness:** 2
**Presentation:** 2
**Contribution:** 2
**Rating:** 2
**Confidence:** 4

**Summary:**

This paper proposes NBF, a NeRF-like model that learns EEG/MEG as a continuous function and renders virtual electrodes. The approach reduces interpolation error and shows accuracy gains for speech decoding and emotion recognition. The ablation results support the stated architectural choices. The main concerns are computational cost and the need for broader validation across datasets and tasks.

**Strengths:**

The approach reduces interpolation error relative to classical interpolators and yields small to moderate gains on downstream tasks such as speech decoding and emotion recognition. Ablations support the importance of positional encoding, normalization choices, skip connections, and progressive initialization. The idea is interesting and potentially impactful.

**Weaknesses:**

The study has potential leakage risks due to per-subject training on sequential 3-second windows with progressive fine-tuning, and it is unclear if normalization, statistics, and hyperparameter search were confined strictly to training electrodes/windows in each split. Baseline coverage is limited, since there is no direct comparison to learning-based virtual-electrode super-resolution methods such as CNN upsampling or GAN approaches that the paper discusses. Electrode geometry and referencing are insufficiently specified, including whether 3-D digitizations or template montages are used, how references are set, and how montage alignment differs across datasets, all of which can materially influence interpolation difficulty and reported metrics.

**Questions:**

- Please specify the spatial referencing schemes used (e.g., average reference, linked mastoids, REST, etc) and analyze how each choice might bias the learned continuous field or interact with the positional encoding used by NBF. Discuss whether NBF can be trained or adapted in a reference-agnostic manner.

- Elaborate the normalization procedures. For z-score, clarify the axis and scope (per channel per trial, per subject, global) and whether statistics are computed on training data only. For min-max, state the range, drift handling, and leakage prevention across splits.

- Discuss cross-subject generalizability. Can NBF align subjects with different montages or sampling grids. Describe any coordinate system, head model, or registration procedure used, and evaluate performance when training on one montage and testing on another.

- The related work and comparisons should include CNN-based virtual electrodes (e.g., Svantesson et al.) and GAN-based super-resolution methods that are already discussed in the text. Please add quantitative comparisons or justify their omission with clear constraints.


Minor corrections:
“NeRFMildenhall et al. (2021)” → “NeRF (Mildenhall et al., 2021)”
Remove the duplicate “EEGNet 2018a/2018b”
“(EasyChair, 2023")” → “(Torma & Szegletes, 2023)”

---

> ### Author Response · Authors · 2025-11-24
> **Answer to Question 1- 2**
>
> 1. In all experiments we trained NBF on the referenced EEG signals exactly as released by each dataset. Because referencing is a linear transform applied uniformly across channels, NBF simply learns the spatial pattern present in those voltages.
> Importantly, NBF does not rely on any specific reference scheme. The positional encoding uses only the electrode coordinates, and the model is invariant to global shifts introduced by different references. This means NBF can be trained on any referenced input as long as the baselines use the same signals.
> NBF does not depend on a particular referencing scheme and can be trained with any referenced input. If desired, one can enforce this explicitly by applying simple preprocessing such as per sample centering or by training on multiple differently referenced versions of the same data, without changing the architecture.
>
> 2.
> Normalization in the data generation pipeline
> In the data generation stage, the model processes the recording in independent three second windows, without a train and validation split inside each window. Normalization is therefore performed entirely within each window and uses only the data from that window.
> Z score normalization of voltage is computed over all electrode samples of the current window. The mean and scale are derived only from that window’s voltage values and are reused to convert model outputs back to voltage. Since the computation is restricted to the window being processed, there is no leakage from other windows or later parts of the recording.
> Coordinate and time values are normalized using window specific minimum and maximum values. Spatial coordinates are mapped into a fixed range from minus one to one, and time is mapped into zero to one. Because the bounds are recomputed for every window, each window effectively defines its own local coordinate frame. Slow changes in the global timestamp or in the coordinate frame do not affect the model because each window is normalized independently. This automatically removes global drift and keeps the information within window scale.
> This procedure ensures that the data generation step remains self contained and avoids any unintended propagation of global information across windows.
>
> Normalization in the interpolation comparison pipeline
> In the interpolation comparison experiments we explicitly separate training and validation electrodes. To prevent leakage, all normalization parameters are derived exclusively from the training electrodes of the current window.
> Z score normalization is computed from voltage values of the training electrodes only. The same training statistics are then applied to the validation electrodes. This ensures that the validation samples cannot influence the normalization procedure and preserves a proper train validation separation.
> Coordinate and time normalization also use minimum and maximum values taken only from training electrodes. The validation electrodes are mapped using these same training bounds. Spatial coordinates are again placed into the fixed range from minus one to one and time into zero to one. Because normalization is based solely on the training subset, the validation predictions cannot leak information back into the training distribution.
> This design prevents both intra window leakage and any contamination across multiple windows. Each window performs its own independent normalization using only its local training samples, preserving experimental integrity for fair comparison with spherical spline and other interpolation baselines.

---

> > ### Author Response · Authors · 2025-11-24
> > **Answers to Questions 3 - 4**
> >
> > 3. Our method is not designed to align subjects across different montages or sampling grids, nor to learn a subject invariant representation. Instead, similar to the design logic used in neural radiance fields, NBF intentionally adopts a per subject modelling strategy that focuses on overfitting a single spatiotemporal “scene.”
> > This reflects a core assumption supported in the EEG literature that each brain exhibits its own spatial organization, conductivity profile, and montage specific distortions, which makes global alignment across subjects inherently ill-posed without strong anatomical constraints.
> > Accordingly, NBF does not employ any head registration procedure, template coordinate system, or cross subject spatial normalization. Each model is trained directly in the native coordinate system of the subject’s own montage, using only that subject’s sensor locations and voltages. Because the model is explicitly designed to learn a dense continuous field for one subject at a time, evaluating performance by training on one montage and testing on another would not be meaningful. Such an experiment would violate the intended behaviour of a neural field model, which, by construction, fits the geometry and signal distribution of its specific recording context.
> > We will revise the manuscript to make this design choice explicit. NBF does not attempt cross subject alignment; rather, it provides a high resolution continuous representation tailored to each subject’s unique montage and signal characteristics. This individualized modelling is central to the method’s purpose: generating spatially coherent EEG estimates at unobserved locations within a single recording.
> >
> > 4. We initially attempted to compare our approach with the recently proposed Spatio Temporal Adaptive Diffusion (STAD) model, which pioneered the application of diffusion models for EEG super resolution on the Localize MI dataset. This dataset comprises high density EEG recordings from seven drug resistant epilepsy patients, acquired using a 256 channel EGI system at 8000Hz. STAD evaluated their method against several deep learning baselines: a convolutional model (DeepCNN), two autoencoder variants and DAE, and a GAN based super resolution method (EEGSR-GAN). The table below presents performance metrics compared to what they reported. While our method demonstrates superior performance across all metrics compared to STAD and the other baseline methods, we lacked access to their specific train/test split and preprocessing details. We contacted the STAD authors to request clarification on their experimental setup, including the exact data segments and train/validation/test partitions, but did not receive a response. Consequently, this comparison was not included in our final evaluation to maintain experimental rigor and reproducibility.
> >
> >
> > **Table 1.** Comparison to prior deep learning based super resolution methods.
> >
> >
> > | Method   | PCC (higher is better) | MAE (lower is better) | NMSE (lower is better) | SNR (higher is better) |
> > |---------|------------------------|------------------------|------------------------|------------------------|
> > | DeepCNN | $0.20 \pm 0.06$        | $0.36 \pm 0.03$        | $0.45 \pm 0.03$        | $3.00 \pm 0.58$        |
> > | WGLAE   | $0.25 \pm 0.06$        | $0.38 \pm 0.03$        | $0.45 \pm 0.03$        | $4.50 \pm 0.29$        |
> > | DAE     | $0.225 \pm 0.06$       | $0.375 \pm 0.03$       | $0.45 \pm 0.03$        | $3.75 \pm 0.29$        |
> > | EEGSR GAN | $0.245 \pm 0.07$     | $0.35 \pm 0.06$        | $0.425 \pm 0.06$       | $4.00 \pm 0.29$        |
> > | STAD    | $0.50 \pm 0.12$        | $0.15 \pm 0.03$        | $0.25 \pm 0.03$        | $8.00 \pm 0.28$        |
> > | Ours    | $0.75 \pm 0.06$        | $0.135 \pm 0.06$       | $0.10 \pm 0.06$        | $10.25 \pm 0.43$       |

---

### Meta-Review · Area_Chair_mBwz · 2025-12-19

**Summary:**

The paper introduces the Neural Brain Fields (NBF) framework, leveraging NeRF-inspired methods for EEG super resolution by modeling scalp voltages as a continuous implicit neural field. The approach claims to improve spatial resolution and handle missing electrode data, showing promising results in speech decoding and emotion recognition. However, significant concerns were raised by reviewers regarding the method's experimental validation, scalability, and the conceptual validity of the NeRF analogy for EEG data.

**Reviewer Concerns:**

Reviewer UoCi:

Addressed by Rebuttal: The authors clarified that NBF is reference-agnostic, and the model can be trained on any referenced input. They also elaborated on the normalization procedures, detailing how normalization is done within each window and ensuring that there is no leakage from other windows.

Outstanding: Despite the clarifications, the concern regarding the lack of comparison with modern learning-based super-resolution methods such as CNN upsampling or GAN approaches was not fully addressed. Additionally, the reviewer raised the issue of generalizability across datasets and tasks, which remains unaddressed in the rebuttal.

Reviewer hoBu:

Addressed by Rebuttal: The authors clarified that the model treats boundary electrodes similarly to other electrodes, with the positional encoding providing stability near the edges. They also explained that NBF does not require cross-subject alignment and works for each subject individually, thus addressing some concerns about generalization.

Outstanding: The concern about the scalability of the model, especially for large datasets or cohorts, was not fully addressed. The rebuttal did not provide a concrete solution to the scalability issues raised by the reviewer. Additionally, the need for further validation in larger or more heterogeneous datasets remains unaddressed.

Reviewer Drmd:

Addressed by Rebuttal: The authors clarified that the model can predict EEG signals at arbitrary time points, explaining that this feature is inherent to the continuous representation but was not evaluated in this submission. They also explained the electrode augmentation process and why specific electrodes were chosen.

Outstanding: The reviewer raised concerns about the analogy between NeRF and EEG, stating that EEG data does not have the same characteristics as NeRF data (e.g., spatial continuity and multi-view constraints). The rebuttal did not adequately address this physical concern or provide further justification for the analogy. Additionally, the scalability issue remains unaddressed, as the rebuttal does not offer a solution to the practicality of training a separate model for each EEG sample.

Reviewer 77z2:

Addressed by Rebuttal: The authors clarified that the model does not rely on extrapolation beyond observed regions and that the boundary electrodes are treated similarly to other electrodes. They also provided additional context on their comparison to the STAD model and noted that they were unable to obtain necessary experimental details for a direct comparison.

Outstanding: The concern that the method is more of an implicit neural representation (INR) rather than a true NeRF was not adequately addressed. The rebuttal did not clarify how the proposed method could align more closely with the NeRF paradigm or offer a more rigorous justification for the physical validity of the model. Furthermore, the reviewer’s concerns about the model’s scalability, especially for large datasets, were not addressed in the rebuttal.

**Reviewer Scores:**

Reviewer UoCi: The score of 2 (reject) would likely remain unchanged due to insufficient experimental validation and a limited baseline comparison, despite the authors’ clarifications on the referencing scheme.

Reviewer hoBu: The score of 8 (accept) may stay, considering the need for broader validation and additional experimental details to confirm the claims.

Reviewer Drmd: The score of 4 (reject, not good enough) would likely stay the same due to the concerns regarding scalability and experimental validation.

Reviewer 77z2: Given the significant concerns about the NeRF analogy and experimental coverage, the score of 2 (reject) is expected to remain unchanged.

---

### Decision · Program_Chairs · 2026-01-26

Reject